# Knowledge of Health Professionals Regarding Vegetarian Diets from Pregnancy to Adolescence: An Observational Study

**DOI:** 10.3390/nu11051149

**Published:** 2019-05-23

**Authors:** Maria Enrica Bettinelli, Elena Bezze, Laura Morasca, Laura Plevani, Gabriele Sorrentino, Daniela Morniroli, Maria Lorella Giannì, Fabio Mosca

**Affiliations:** 1Department of Clinical Sciences and Community Health, University of Milan, via San Barnaba 8, 20122 Milan, Italy; maria.bettinelli@unimi.it (M.E.B.); laura.morasca@94gmail.com (L.M.); fabio.mosca@unimi.it (F.M.); 2Fondazione IRCCS Cà Granda Ospedale Maggiore Policlinico, via Commenda 12, 20122 Milan, Italy; elena.bezze@policlinico.mi.it (E.B.); laura.plevani@mangiagalli.it (L.P.); gabriele.sorrentino@policlinico.mi.it (G.S.); daniela.morniroli@gmail.com (D.M.)

**Keywords:** vegetarian, diet, health professional, knowledge, attitude, food model

## Abstract

The number of people adopting vegetarian diets is constantly increasing, and many among them are young parents who decide to share their diet with their children. The aim of this study was to investigate health professionals’ knowledge regarding the adoption of vegetarian diets from pregnancy to adolescence. A cross-sectional survey was conducted. The administered questionnaire, which was based on the recommendations of the most up-to-date guidelines, included two macro areas: The first investigated the sociodemographic and professional profile of the interviewees and the second addressed the knowledge of the participants regarding vegetarian diets. A total of 418 health professionals in Italy were interviewed, of whom 65.8% were nursing staff. Among the participants, 79.9% had not attended a nutrition course in the previous five years. A correct definition of a vegetarian/vegan diet was provided by 34.1% of the participants. The answers regarding knowledge of nutrients were correct in 20% of cases, whereas correct answers to questions assessing knowledge of the risk and benefits of a vegetarian diet and the adoption of a vegetarian diet throughout the life cycle were given by 45% and 39.4% of the participants, respectively. A significant correlation between the items of the second macro area that investigated the knowledge and dietary habits of the participants was found for seven items. The results of the study indicate that health professionals do not have complete and exhaustive knowledge about vegetarian diets and lack information on health outcomes and the adoption of a vegetarian diet throughout the different life cycles and nutrients. Improving pre- and in-service learning opportunities in vegetarian nutrition for health professionals is strongly advisable.

## 1. Introduction

Appropriate and adequate nutrition in early life is strictly interrelated with subsequent health outcomes [1,2]. Several health benefits have been associated with the adoption of well-planned vegetarian-type diets, including protection from metabolic and cardiovascular diseases and some types of neoplasia [3,4]. 

Vegetarian diets are characterized by different nutritional profiles depending on the degree of restriction or exclusion of one or more types of food. The most common dietary models are the lacto-ovo vegetarian diet (excludes meat and fish but includes dairy products, eggs, and honey, together with a wide variety of plant foods), the lacto-vegetarian diet (excludes meat, fish, and eggs but includes dairy products and honey, together with a wide variety of plant foods), the ovo-vegetarian diet (excludes meat, fish, and dairy products but includes eggs and honey, together with a wide variety of plant foods), and the vegan diet (excludes meat, fish, dairy products, eggs, and honey but includes a wide variety of plant foods) [3].

Concern has arisen regarding possible nutrient deficiencies, such as iron, zinc, vitamin D, vitamin B12, iodine, proteins, and *n*-3 fatty acids, especially when vegetarian diets are consumed during pregnancy or during the growth phase [5]. 

The inadequacy of vegetarian diets not appropriately planned and balanced has been underlined by several scientific societies, particularly with regard to the promotion of children’s neuropsychomotor development [6]. The occurrence of failure to thrive, neurodevelopmental regression, and megaloblastic anemia due to vitamin B12 deficiency have been reported in infants adopting strict vegetarian diets and/or with limited access to foods of animal origin [7]. 

The European Society for Paediatric Gastroenterology Hepatology and Nutrition (ESPGHAN) [8] recommends a careful evaluation of potential nutritional deficiencies due to the severe neurological consequences associated with the lack of adequate nutritional supplementation. The Academy of Nutrition and Dietetics [9], Canada’s Food Guide, the American Dietetic Association, Dietitians of Canada, the American Academy of Pediatrics, and the Canadian Paediatric Society [4,10] have taken favorable position regarding vegetarian diets at all stages of life, including pregnancy, lactation, infancy, childhood, and adolescence, as long as appropriate attention to critical nutrients is paid.

The number of people who are adopting a vegetarian lifestyle is constantly increasing; it is estimated at approximately 5% and 3.3% of the populations in Europe [11] and the United States [9], respectively. According to the latest data from the EURISPES Italy 2018 Report, 7% of the Italian population aged 18 years and over follow a vegetarian diet, and 0.9% of this group report following a vegan diet [12]. Many vegetarians are young parents who decide to share their dietary models with their children [12]. In this context, a potential risk is that people will adopt a diet because it is fashionable or due to ethical issues related to animal welfare [13,14], without being adequately educated and informed of the potentially negative health effects of inappropriately planned diets.

It is of utmost importance that health professionals who care for pregnant women, newborns, and children are able to provide proper nutritional education for their patients in order to ensure the adequate intake of all nutrients and avoid nutritional imbalances [15,16]. Indeed, meeting nutritional needs during critical periods in early life is crucial for the developmental programming of later health outcomes. A well-documented example of this importance is the neurological unfavorable outcome of children born to mothers with severe vitamin B12 deficiency [17]. Vitamin B12 deficiency has also been linked to long-term neurological disturbances such as an increased risk of adulthood depression in babies born to vegan or vegetarian mothers. These outcomes describe the significant nexus between nutrition during pregnancy and fetal programming [18]. Moreover, adoption of well-planned vegetarian-type diets may represent an additional useful tool in reducing the economic and health burden of metabolic syndrome [3].

There is a paucity of studies addressing health professionals’ knowledge regarding vegetarian nutrition. The aim of this study was to investigate health professionals’ level of knowledge regarding the impacts of vegetarian diets on fetal, infant, and child development.

## 2. Materials and Methods

A cross-sectional study was conducted that included a convenience sample of health professionals working in the maternal–child area of two tertiary level hospitals in the metropolitan area in Milan. The Ethics Committee of the “Fondazione Istituto di Ricovero e Cura a Carattere Scientifico Cà Granda Ospedale Maggiore Policlinico” approved the study, and written informed consent was obtained from the participants.

Nurses, pediatric nurses, midwives, staff nurses, and health care support workers were included. Nonmedical health professionals, medical doctors, and administrative staff were excluded. 

A questionnaire was composed according to the recommendations of the most up-to-date and authoritative scientific societies and guidelines [1,10,14,19,20]. Prior to the start of the study, the questionnaire was tested by a group of nurses, who were then excluded from the survey, to clarify any doubts regarding the comprehension of the items. The questionnaire, shown in the Appendix A, consisted of 36 multiple choice questions subdivided into 2 macro areas. The first macro area included 13 questions: 9 concerning the professionals’ sociodemographic and professional profiles and 4 related to their eating behaviors. The second macro area consisted of 23 questions and aimed to verify the interviewed health professionals’ knowledge regarding four specific subtopics. Specifically, the questions addressed the definition of a vegetarian/vegan diet (subtopic 1: Questions 1 and 2), knowledge regarding the risks and benefits associated with such diets (subtopic 2: Questions 3 to 5), knowledge regarding specific nutrients (subtopic 3: Questions 17 to 23), and the adoption of a vegetarian diet throughout various life cycle stages, including pregnancy, the first and second phases of childhood, and adolescence (subtopic 4: questions 6 to 16). The participants’ answers were considered correct when they were consistent with the definitions provided by scientific societies and guidelines [1,7,10,14,19,21,22,23].

The participants were classified as omnivores, semivegetarians, and vegetarians according to the definitions of Lap Tai Le and Joan Sabatè (Table 1) [3].

Data were collected between 1 August and 30 September, 2017. The investigators in charge of the study were responsible for explaining the questionnaire through a face-to-face interview with the enrolled health professionals. The questionnaire was then self-administered, and nearly 15 minutes were needed to complete it. 

### Statistical Analysis

Data were expressed as absolute and percentage of observations. Associations between the 23 questionnaire items investigating the participants’ knowledge, sociodemographic and professional characteristics, and dietary patterns were assessed using the chi-square test or the “Fisher’s test” when applicable.

A value of P ≤ 0.05 was considered statistically significant. The statistical analyses were performed using SPSS version 12 statistic software package (SPSS Inc., Chicago, IL, USA).

## 3. Results

Of the 611 eligible health professionals, 40 (6.6%) were on vacation. Moreover, 193 (31.6%) health professionals were not reachable during the day shift. As a result, 418 (68.4%) health professionals entered the study. The nursing personnel completed 275 (65.8%) questionnaires and were equally distributed by professional profile: 140 nurses (33.5%) and 135 pediatric nurses (32.3%). The other participants were as follows: 60 (14.4%) midwives, 43 (10.3%) health care support workers, and 40 (9.6%) staff nurses. Table 2 shows the basic characteristics of the sample. Most of the participants were female, Italian, and had a college degree. A total of 61% of the participants were older than 39 years. Half of the participants had work experience equal to or more than 20 years, with the neonatal and pediatric areas being the most represented.

According to the answers concerning the frequency of food intake, 342 (77.5%) participants were classified as omnivores, 73 (17.5%) as semivegetarians, and 21 (5%) as vegetarians.

Table 3 shows the absolute and percentage of correct answers given to the items investigating the participants’ knowledge. The results are reported according to the different subtopics addressed in the second macro area of the questionnaire. A correct definition of a vegetarian and a vegan diet was provided by 2.2% and 66% of the participants, respectively. For the subtopics “Risk and Benefits of a Vegetarian/Vegan Diet” and the “Effects of a Vegetarian Diet Throughout the Life Cycle”, the participants scored the highest number of correct answers (45.0% and 39.4%, respectively), whereas their answers concerning nutrients were correct in one out of five cases (20.7%).

No significant association was found between the nutrition training in the previous 5 years, the sociodemographic and professional characteristics of the sample, and the answers they gave to the 23 items on the second macro area of the questionnaire.

A significant correlation was found between 7 of the 23 items on the second macro area investigating the participants’ knowledge and dietary habits according to Lap Tai Le and Joan Sabatè (omnivorous, semivegetarian, and vegetarian) [3] (Table 4). Regarding the first 5 items reported in Table 4, the participants who were classified as having a vegetarian dietary pattern gave correct answers in a higher percentage of cases than those classified as semivegetarian or omnivorous. In relation to the item investigating whether a plant-based diet can satisfy the nutritional demands of an infant in the first 1000 days of life, the percentage of correct answers provided by omnivores was higher than that provided by vegetarians. When considering the item “a vegan mother can breastfeed without supplementation,” the percentage of correct answers was similar among omnivores and vegetarians, with the semivegetarians providing the lowest percentage of correct answers.

## 4. Discussion

To the best of our knowledge, this is the first study addressing the knowledge of health professionals working in maternal–child areas within a hospital setting. The results of the present study indicate that the interviewed health professionals do not have complete and exhaustive knowledge regarding vegetarian diets. For the first subtopic concerning the definition of vegetarian diet (excludes all types of meat, fish but may include dairy products, eggs, and honey) and vegan diet (excludes meat, fish, dairy products, eggs, and honey), the most common variants of the definition were not identified correctly. Moreover, the results of the study suggest a lack of information regarding health outcomes and nutrients. With regard to adherence to vegetarian diets throughout the life cycle, from pregnancy to adolescence, the interviewed health professionals demonstrated a general lack of knowledge. These findings could be at least partially explained by the fact that only a limited percentage of the study participants had taken staff nutrition courses in the previous 5 years, highlighting the importance of continuing education to address the rapid changes in healthcare and in practice to achieve improved outcomes. Moreover, in view of the strict interrelationship between early nutrition and health outcomes, it is extremely important that health care professionals have adequate knowledge of these topics to correctly guide and educate patients who adopt vegetarian diets during the critical and plastic phases of growth and development. The crucial role of early nutrition in mediating the programming of metabolic health in later life has been extensively documented [20]. Accordingly, increasing evidence indicates the importance of developing an interprofessional approach when providing nutritional care throughout the different life stages [24].

However, it has to be taken into consideration that, in the present study, no significant relationship was found between the sociodemographic and professional characteristics, including the attendance to specific training and the level of health professionals’ knowledge, probably because the study lacked adequate statistical power to find it. On the other hand, it can also be hypothesized that results could have been biased by the personal beliefs of the enrolled health professionals on their dietary choice rather than merely reflecting the acquired specific knowledge. 

International and national scientific societies focus on proper diet planning and integration to avoid nutritional imbalances. The importance of adequate supplementation with vitamin B12 was particularly emphasized due to the serious clinical consequences associated with its deficiency [17,18,25]. Among the interviewed personnel, more than half of the sample (55.7%) correctly identified this nutrient as the one for which that vegetarian/vegan mothers had the greatest risk of deficiency during pregnancy.

According to the results of this study, the answers given for the 4 subtopics addressing knowledge of vegetarian diets were not relevantly associated with the participants’ adopted diet, with the exception of seveb items. In particular, most of the health professionals who adopted a vegetarian diet appeared to have good knowledge of the major benefits of a plant-based diet and the adequacy of a planned vegetarian diet during various stages of the life cycle. These results are consistent with previous findings indicating that attitudes towards a vegetarian lifestyle are significantly correlated with nutritional knowledge [26]. However, it must also be taken into account that when the health professionals were asked whether a plant-based diet could meet the nutritional demands of infants in the first 1000 days of life, the percentage of correct answers given by the health professionals who adopted an omnivorous diet was actually higher than the percentage of correct answers selected by the professionals who followed a vegetarian diet. Hence, it can be speculated that even vegetarians can be misinformed about the concept of planning, which is essential to an adequate vegetarian diet.

This study confirms the current perceived barriers to adopting a vegetarian diet, as shown by Corrin et al. [26]. The authors reported the perception that vegetarian diets are nutritionally unbalanced, the existing lack of information about diets, and a general worry about overall health. Although there is increasing interest in vegetarianism, nutrition education programs and food-based dietary guidelines should be promoted in view of the strong link between diet and health outcomes and the increasing evidence of the health and environmental benefits associated with a diet rich in plant-based food and with fewer animal source foods [27].

To our knowledge, the present study is the first to offer insights into the knowledge of a relatively high number of health professionals. This study has some limitations since the collected data regarding the health professionals’ level of knowledge refer to a convenience sample and the results may not be generalizable. Moreover, no data concerning medical staff and personal beliefs on dietary choices were collected.

## 5. Conclusions

Health professionals do not have complete and exhaustive knowledge regarding vegetarian diets, and they lack information on health outcomes, the adoption of the vegetarian diet during different stages of the life cycle, and nutrients.

Pre- and in-service education programs should be improved to ensure adequate knowledge of vegetarian nutrition, therefore enabling health professionals to provide appropriate educational intervention and guidance, and detect nutritional imbalances that, if not identified in a timely manner, can lead to serious consequences.

## Figures and Tables

**Table 1 nutrients-11-01149-t001:** Classification of dietary patterns.

Dietary Pattern	Definition
Omnivorous	Eating red meat, poultry, fish, milk, and eggs more than once a week
Semivegetarian	Eating red meat, poultry, and fish less than once a week and more than once a month
Vegetarian	
Pesco	Eating fish, milk, and eggs, but no red meat or poultry
Lacto-ovo	Eating milk, eggs, or both but no red meat, poultry, or fish
Vegan	Eating no red meat, poultry, fish, dairy, or eggs

**Table 2 nutrients-11-01149-t002:** Population characteristics.

Sociodemographic Characteristics	Variable	N	%
Gender	Female	377	90.2
	Male	41	9.8
Nationality	Italian	411	98.3
	Non-Italian	7	1.7
Age (years)	<30	55	13.2
	30–39	105	25.0
	40–49	128	30.7
	>50	130	31.1
Education	High school diploma	83	19.8
	College degree	335	80.2
**Professional characteristics**			
Work experience (years)	<9	127	30.4
	10–19	69	16.5
	20–29	135	32.3
	>30	87	20.8
Work experience (areas)	Obstetrics	91	21.8
	Neonatal	113	27.0
	Pediatrics	143	34.2
	Community service	1	0.2
	Two or more areas	70	16.8
Participation in a nutrition course in the last 5 years	Yes	84	20.1
	No	334	79.9
Asked by parents for information on their child’s feeding	Always, often	174	41.6
	Never, rarely	244	58.4

**Table 3 nutrients-11-01149-t003:** Absolute and percentage correct answers regarding knowledge about vegetarian diets. Correct answers determined by definitions provided by scientific societies and guidelines [1,7,10,14,19,21,22,23].

**Subtopic 1: Definition of a Vegetarian/Vegan Diet**		
**Items**	**Correct Answer**	**%**
1.Vegetarian diet	Exclusion of meat, fish and poultry + eggs and/or milk and dairy products	2.2
2.Vegan diet	Exclusion of meat, fish and poultry, eggs, dairy products, and honey	66.0
**Subtopic 2: Risk and Benefits of a Vegetarian/Vegan Diet**		
**Items**	**Correct Answer**	**%**
3.An animal-based diet provides more health benefits	False	55.3
4.A vegan diet does not further reduce the risks of cardiovascular disease and diabetes	False	32.1
5.Many chronic diseases are…	More common in omnivores than in vegetarians	47.6
**Subtopic 3: Nutrients**		
**Items**	**Correct Answer**	**%**
17.Plant proteins have less bioavailability	True	16.3
18.In the first 2 years of life, the amount of fiber must be limited	True	16.5
19.Iron is a critical nutrient only for vegetarian/vegan children under 2 years of age	False	55.8
20.The sources that meet the calcium requirements are…	Breast milk, formula, calcium low-salt water	4.1
21.Zinc has…	Greater bioavailability in an omnivorous diet	7.2
22.Vitamin D levels appear to be influenced by diet and are lower in vegetarian children	False	30.6
23.Long-chain omega-3 fatty acid levels are…	Lower in vegetarians, typically absent in vegans	14.1
**Subtopic 4: Effects of a Vegetarian Diet Throughout the Life Cycle**		
**Items**	**Correct Answer**	**%**
6.A planned vegetarian diet is nutritionally adequate during all stages of the life cycle	True	31.1
7.A plant-based diet is able to satisfy the nutritional demands of an infant in the first 1000 days of life	False	61.2
8.The vegetarian/vegan mother is at greater risk of deficiency in…	Vitamin B12	55.7
9.Planned vegetarian and vegan diets during pregnancy present higher risks of pregnancy difficulties and birth defects	False	33.7
10.Lacto-ovo vegetarian mothers can breastfeed without supplementation	True	34.7
11.Vegan mothers can breastfeed without supplementation	False	51.7
12.Children who follow a planned vegetarian diet have an adequate energy intake similar to that of omnivorous children	True	32.3
13.During complementary feeding, the infant must receive a sufficient amount of…	Vitamin B12, vitamin D, iron, zinc, folate, *n*-3 LCPUFA, protein and calcium	57.9
14.A vegetarian/vegan diet is a protective factor against infant overweight	True	30.6
15.The supplemental nutrient that should not be missing for a vegetarian/vegan child is…	Vitamin B12	40.4
16.A teenager who chooses a vegetarian diet is…	Potentially at risk for making impulsive choices, ill-informed choices, and having an eating disorder	4.5

**Table 4 nutrients-11-01149-t004:** Significant relationships between the interviewees’ survey answers and their dietary habits.

Item	Correct Answer		%		
			Dietary Pattern		
		Omnivorous	Semi-Vegetarian	Vegetarian	*P**
An animal-based diet provides more health benefits	False	50.9	64.4	90.5	0.003
A planned vegetarian diet is nutritionally adequate during all stages of the life cycle	True	24.1	45.2	90.5	0.000
Planned vegetarian and vegan diets during pregnancy involve higher risks of pregnancy difficulties or birth defects	False	30.6	35.6	76.2	0.001
Lacto-ovo vegetarian mothers can breastfeed without supplementation	True	30.6	46.6	57.1	0.006
Children who follow a planned vegetarian diet have an adequate energy intake similar to that of omnivorous children	True	26.9	46.6	66.7	0.000
Vegan mothers can breastfeed without supplementation	False	52.8	46.6	52.4	0.019
A plant-based diet is able to satisfy the nutritional demands of an infant in the first 1000 days of life	False	64.8	52.1	38.1	0.000

* *P* ≤ 0.05, statistically significant.

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
