# Peer review of "Knowledge of Health Professionals Regarding Vegetarian Diets from Pregnancy to Adolescence: An Observational Study"

_nutrients, 2019, doi:10.3390/nu11051149_

Reviewer 1 Report

Review of “Knowledge of health professionals regarding vegetarian diets from pregnancy to adolescence: an observational study” by Bettinelli et al. Nutrients, April 2019.

The manuscript prepared by Bettinelli et al. provides results from an interesting preliminary study examining the accuracy of identification of a vegetarian or vegan diet through a series of simple questionnaires. Overall the manuscript is interesting and demonstrates that health care practitioners overall do not have a strong understanding of nutrition when related to the basic differences between omnivore, vegetarian, and vegan diets for use at different stages of life. It will be interesting to see if the education provided in those practitioners who have attended training in the last 5 years is sufficient or if major changes need to be made to this area of the medical education system in Italy, and perhaps elsewhere. This novel study, highlighted by the lack of previous publications of its kind, could be improved using their readily available data to determine if these faults lie within the practitioner’s lack of recent education, the lack of accurate education being provided, or the personal bias inherent in our beliefs on diet that are not supported by research findings. 

General Comments:

1.      This is a very interesting and hot topic at the moment but the introduction feels repetitive; the third paragraph especially so. This is likely due to a lack of detail on the subject matter leading to the introduction seeming vague. If the authors added perhaps a few examples of which nutritional deficiencies are the most dramatic for human health and what it is that is most clearly different between a strict vegetarian diet vs a traditional diet I feel it would make the introduction more interesting and impactful. As it stands, I feel I need to go back and read every reference just to understand the differences when this could be clearly explained in a few short sentences. This is the punch line of the study, make sure you utilise it.

2.      The authors results highlight that while practitioners who are themselves vegetarian appear to score more accurately on the questions suggesting the safety or benefits of a vegetarian diet, they do not discuss the potential personal bias in which individuals may think that their own dietary decisions are simply better than others. In examining the responses in table 4, the reviewer wonders how much of the response is related to an accurate knowledge of nutrient values vs bias towards one’s own beliefs in their dietary decisions. If you follow through table 4 with this hypothesis in mind, you can see a trend for the answer accuracy of vegetarians through the questions. For example:

A.     Animal-based diet provides more health benefits; of course not, my vegetarian diet is the better choice. Vegetarians scored accurately on this.

B.     A planned vegetarian diet is nutritionally adequate; of course it is, my vegetarian diet is the better choice. Vegetarians scored accurately on this.

C.      A plant based diet is able to satisfy the nutritional demand of an infant; of course it can, my vegetarian beliefs are a great choice. Vegetarians biasedly scored inaccurately here.

While this is merely an alternative hypothesis to the assumption that some practitioners are more educated than others, I believe that it is worth postulating that personal bias may have a great deal to do with the specific answers these individuals provided.

3.      The authors indicate in the discussion that their “findings could be at least partially explained by the fact that only a limited percentage of the study participants had taken staff nutrition courses in the previous 5 years” however they do not make any attempt to correlate these findings. The study would be made much more impactful if the authors used their readily available data to demonstrate that those health care professionals who DID have nutrition training in the last 5 years (accounting for 20% of the cohort) were significantly more accurate on their questionnaires than those without. Please include this statistical analysis in the manuscript.

Specific Comments:

1.      Table 3 contains no references for any peer reviewed studies to support the correct answers to the questions provided. References must be provided as these data are not supported by results from the study at hand.

2.      Table 3 questions 7 and 12 seem equivalent, yet the answer is different. If a plant-based diet is not able to provide nutrition to an infant in the first 1000 days, and a planned vegetarian diet is equivalent to the energy intake of an omnivorous diet in children then this would indicate that no child receives adequate nutrition in the first 1000 days. As the definition of infant and child, which the reviewer can only assume in this study transitions at 1000 days, is not included prior to this point, the difference remains unclear. This is not mentioned until line 155, please ensure you define these two cohort earlier with reference to indicate why this specific time point was chosen.    

Author Response

Review Report Form N.1 

Review of “Knowledge of health professionals regarding vegetarian diets from pregnancy to adolescence: an observational study” by Bettinelli et al. Nutrients, April 2019.

The manuscript prepared by Bettinelli et al. provides results from an interesting preliminary study examining the accuracy of identification of a vegetarian or vegan diet through a series of simple questionnaires. Overall the manuscript is interesting and demonstrates that health care practitioners overall do not have a strong understanding of nutrition when related to the basic differences between omnivore, vegetarian, and vegan diets for use at different stages of life. It will be interesting to see if the education provided in those practitioners who have attended training in the last 5 years is sufficient or if major changes need to be made to this area of the medical education system in Italy, and perhaps elsewhere. 

We thank you the reviewer for the comments. We believe the results of this study will be helpful in planning an implementation of health care professionals’ education, including doctors, both during the pre-service and the in-service training. 

This novel study, highlighted by the lack of previous publications of its kind, could be improved using their readily available data to determine if these faults lie within the practitioner’s lack of recent education, the lack of accurate education being provided, or the personal bias inherent in our beliefs on diet that are not supported by research findings. 

Thank you very much for this interesting suggestion. We agree with the reviewer that personal beliefs on dietary choice could affect level of knowledge. Unfortunately, we have not assessed personal beliefs in this study but, when planning a larger one, we will definitely assess them. 

General Comments:

1.     This is a very interesting and hot topic at the moment but the introduction feels repetitive; the third paragraph especially so. This is likely due to a lack of detail on the subject matter leading to the introduction seeming vague. If the authors added perhaps a few examples of which nutritional deficiencies are the most dramatic for human health and what it is that is most clearly different between a strict vegetarian diet vs a traditional diet I feel it would make the introduction more interesting and impactful. As it stands, I feel I need to go back and read every reference just to understand the differences when this could be clearly explained in a few short sentences. This is the punch line of the study, make sure you utilise it.

According to your suggestions, we have shortened the introduction, focusing on the main issue of the paper. We have inserted some examples of risks of t not well-planned vegetarian diets and we have detailed the main differences among the different types of vegetarians diets.

2.      The authors results highlight that while practitioners who themselves are vegetarian appear to score more accurately on the questions suggesting the safety or benefits of a vegetarian diet, they do not discuss the potential personal bias in which individuals may think that their own dietary decisions are simply better than others. In examining the responses in table 4, the reviewer wonders how much of the response is related to an accurate knowledge of nutrient values vs bias towards one’s own beliefs in their dietary decisions. If you follow through table 4 with this hypothesis in mind, you can see a trend for the answer accuracy of vegetarians through the questions. For example:

A.     Animal-based diet provides more health benefits; of course not, my vegetarian diet is the better choice. Vegetarians scored accurately on this.

B.     A planned vegetarian diet is nutritionally adequate; of course it is, my vegetarian diet is the better choice. Vegetarians scored accurately on this.

C.      A plant based diet is able to satisfy the nutritional demand of an infant; of course it can, my vegetarian beliefs are a great choice. Vegetarians biasedly scored inaccurately here.

While this is merely an alternative hypothesis to the assumption that some practitioners are more educated than others, I believe that it is worth postulating that personal bias may have a great deal to do with the specific answers these individuals provided.

Thank you for your comments. According to your suggestion, we have considered the potential effect of personal beliefs on dietary choices in the discussion 

3.      The authors indicate in the discussion that their “findings could be at least partially explained by the fact that only a limited percentage of the study participants had taken staff nutrition courses in the previous 5 years” however they do not make any attempt to correlate these findings. The study would be made much more impactful if the authors used their readily available data to demonstrate that those health care professionals who DID have nutrition training in the last 5 years (accounting for 20% of the cohort) were significantly more accurate on their questionnaires than those without. Please include this statistical analysis in the manuscript.

We agree with the reviewer that it is important to try to correlate these findings. Accordingly, we  have actually did this analysis, without finding any significant correlation.  We have explained this point better in the Results and inserted a comment in the discussion. 

Specific Comments:

1.      Table 3 contains no references for any peer reviewed studies to support the correct answers to the questions provided. References must be provided as these data are not supported by results from the study at hand.

Thank you for your suggestion. We have inserted references to support the correct answers.

2.      Table 3 questions 7 and 12 seem equivalent, yet the answer is different. If a plant-based diet is not able to provide nutrition to an infant in the first 1000 days, and a planned vegetarian diet is equivalent to the energy intake of an omnivorous diet in children then this would indicate that no child receives adequate nutrition in the first 1000 days. As the definition of infant and child, which the reviewer can only assume in this study transitions at 1000 days, is not included prior to this point, the difference remains unclear. This is not mentioned until line 155, please ensure you define these two cohort earlier with reference to indicate why this specific time point was chosen. 

We thank you the reviewer for the comment. The differences about the two questions refer to the period (1000 days vs childhood) and to the consumption of a well-planned vegetarian diet vs a plant-based diet. We have chosen to use these two periods to understand if health professionals know the influence of nutrition in the first 1000 days and during childhood and the importance of a well-planned diet. We have detailed this point in the methods section, defining subtopic 4 as related to the adoption of a vegetarian diet throughout various life cycle stages, including pregnancy, the first and second phases of childhood and adolescence. Accordingly, we have used the term “infant” when referring to the first phase of childhood and the term “child” when referring to the second phase.

Reviewer 2 Report

I think the topic of this article is of great importance – it is very central that health professional have an adequate knowledge regarding the impacts of vegetarian diets on fetal, infant and child development. As maternal nutrition during pregnancy is very essential for an adequate fetal development, I would mention the concept of the Developmental Origins of Health and Disease to emphasize the significance of maternal nutrition.

I do not fully understand why the introduction starts with the Mediterranean diet – of course, the Mediterranean diet is a very plant-based diet, but it also contains dairy products, meat, and especially fish. I see the point that the Mediterranean diet goes in the direction of a vegetarian diet, but the Mediterranean diet is definitely not vegetarian/vegan or restrictive. In my opinion, I would start right away with the prevalence of vegetarianism/veganism. Also, I recommend explaining the different dietary models of the vegetarian/vegan diets in the running text, since not all people are familiar with these terms.

Nurses, pediatric nurses, midwifes, staff nurses and health care support workers were included in your study – was there no data available on gynecologists? I am also wondering if you collected data on the weight and height of the health professionals in order to calculate the BMI, which could be an appropriate marker for their own nutritional health.

“Of the 611 eligible health professionals, 193 (31.6%) refused to participate...” – were there any reasons why they refused to participate? If yes, I suggest mentioning reasons as this is quite a large number of refusals.

“Remarkably, a correct definition of a vegetarian/vegan diet was provided by only one-third of the participants (34.1%).” --> I suggest splitting the results into two parts (vegetarian vs. vegan definition) since there is a huge difference (2.2% vs. 66%) between the correct answer percentage among vegetarian and vegan diets.

“Moreover, in view of the strict interrelationship between early nutrition and health outcomes, it is extremely important that health care professionals have adequate knowledge of these topics to correctly guide and educate patients who adopt vegetarian diets during the critical and plastic phases of growth and development…” --> I am wondering why you did not refer to specialized nutritionists or dieticians for nutrition counselling in critical phases like pregnancy or growth. I think that interdisciplinary is very important in the mother-child related context. In my opinion, it is not the duty of a nurse to do nutrition counselling, since they are not experts in this field – it is the task of well trained nutritionists or dieticians. Especially in your conclusion, I would highlight the importance of interdisciplinary.

Author Response

Comments and Suggestions for Authors

I think the topic of this article is of great importance – it is very central that health professional have an adequate knowledge regarding the impacts of vegetarian diets on fetal, infant and child development. As maternal nutrition during pregnancy is very essential for an adequate fetal development, I would mention the concept of the Developmental Origins of Health and Disease to emphasize the significance of maternal nutrition.

We thank you the reviewer for the comments. We have inserted the concept of the Developmental Origins of Health and Disease in the introduction and discussion section.

I do not fully understand why the introduction starts with the Mediterranean diet – of course, the Mediterranean diet is a very plant-based diet, but it also contains dairy products, meat, and especially fish. I see the point that the Mediterranean diet goes in the direction of a vegetarian diet, but the Mediterranean diet is definitely not vegetarian/vegan or restrictive. In my opinion, I would start right away with the prevalence of vegetarianism/veganism. Also, I recommend explaining the different dietary models of the vegetarian/vegan diets in the running text, since not all people are familiar with these terms.

We agree with your comment and modified the introduction as you suggested.

Nurses, pediatric nurses, midwifes, staff nurses and health care support workers were included in your study – was there no data available on gynecologists? 

The survey was carried out only on a sample of nurses, pediatric nurses, midwifes, staff nurses and health care support workers.

I am also wondering if you collected data on the weight and height of the health professionals in order to calculate the BMI, which could be an appropriate marker for their own nutritional health.

Sorry, we didn’t consider this suggestive aspect. We will take in mind it for a further study.

 “Of the 611 eligible health professionals, 193 (31.6%) refused to participate...” – were there any reasons why they refused to participate? If yes, I suggest mentioning reasons as this is quite a large number of refusals.

We apologize for not having been clear with regard to this point. The refusal to participate in the study was actually depending on absence for vacations or difficulty to meet the health workers by interviewer during daily service. We have therefore changed the results in order to make this point clear. 

“Remarkably, a correct definition of a vegetarian/vegan diet was provided by only one-third of the participants (34.1%).” --> I suggest splitting the results into two parts (vegetarian vs. vegan definition) since there is a huge difference (2.2% vs. 66%) between the correct answer percentage among vegetarian and vegan diets.

We agree with your consideration. Accordingly, we have changed the text, splitting vegetarian and vegan. 

“Moreover, in view of the strict interrelationship between early nutrition and health outcomes, it is extremely important that health care professionals have adequate knowledge of these topics to correctly guide and educate patients who adopt vegetarian diets during the critical and plastic phases of growth and development…” --> I am wondering why you did not refer to specialized nutritionists or dieticians for nutrition counselling in critical phases like pregnancy or growth. I think that interdisciplinary is very important in the mother-child related context. In my opinion, it is not the duty of a nurse to do nutrition counselling, since they are not experts in this field – it is the task of well trained nutritionists or dieticians. Especially in your conclusion, I would highlight the importance of interdisciplinary.

We thank you the reviewer for the comment and we agree on the importance of a multidisciplinary approach to patients’ nutritional counselling. We have highlighted its importance in the discussion section. In view of this multidisciplinary approach,  we think that also nurses need to have the necessary nutritional education to collect a nutritional history during pregnancy, lactation and childhood, and contribute to the nutritional counselling to the parents. Obviously, planning the diets is a task of nutritionists and well-trained doctors. 

Reviewer 3 Report

The paper has potential, however 

x) the introduction is insufficient as it does not lead the reader to the main issue

x) the method/statistical analysis is a bit too simple - run again and improve consdering comments

x) presentation of items of questionnaire could be improved, bit confusion/better overview would be preferable

x) discussion is insufficient mainly because of weak statistics, but also as a consequence of insuficcient introduction, it should in length be 2-3times the intro section, with new run stats it could add quality and length, explicitly line out all the limitations, as there are more than 1-2

x) english style could be much improved, native speaker with scientific background should be involved

Author Response

Reviewer 3 

The paper has potential, however 

x) the introduction is insufficient as it does not lead the reader to the main issue

We thank you the reviewer for the helpful comments. We have modified the introduction focusing on the main issue of the paper.

x) the method/statistical analysis is a bit too simple - run again and improve considering comments

We agree with the reviewer that the method/statistical analysis is a bit simple. However, we choose to perform this analysis since we aimed to investigate health professionals’ level of knowledge regarding the impacts of vegetarian diets on foetal, infant and child development. Moreover, we assessed whether there was any association among the level of knowledge and the collected socio-demographics characteristics. However, the number of subjects enrolled in the study, although globally rather large, did not allow to perform additional statistical analysis since the number of the subjects categorized according to the dietary patterns were relatively small especially in some specific subgroups (i.e. vegetarians). 

x) presentation of items of questionnaire could be improved, bit confusion/better overview would be preferable

We agree with the reviewer that table 3 is busy. IN order to make the table easier for the reader we have  made some changes.  

x) discussion is insufficient mainly because of weak statistics, but also as a consequence of insufficient introduction, it should in length be 2-3 times the intro section, with new run stats it could add quality and length, explicitly line out all the limitations, as there are more than 1-2

As stated in the answer to the previous point, the primary aim of the study was to assess the health professionals’ level of knowledge regarding the impacts of vegetarian diets on foetal, infant and child development. Moreover, the number of subjects enrolled in the study, although globally rather large, did not allow to perform additional statistical analysis since the number of the subjects categorized according to the dietary patterns were relatively small especially in some specific subgroups (i.e. vegetarians).

With regard to the discussion, we are aware that is rather short. However, since, to our knowledge, this is the first paper addressing health professionals’ level of knowledge, we could not compare other results with other previous surveys. We have added to the limitation of the study that no data concerning medical staff knowledge has been collected.

x) english style could be much improved, native speaker with scientific background should be involved

The paper has been edited by a professional english editing service before submission. For your convenience, we have attached the certificate stating that the paper has been edited.

Annotations of nutrients-494988-review
> highlight [page 1]: 41 Accordingly, in November 42 2010, UNESCO recognized the Mediterranean diet as “intangible heritage of humanity”. In addition 43 to its recognized health benefits, the Mediterranean diet has also been proposed as a sustainable . 
According to your suggestion, we have deleted this part. Please see the reviewed version.
> note [page 1]: from Italy, other nations? Italian- 
we have specified in the text
> note [page 1]: in which specific settings it is suggested to be highly advised, eg. 1) ... 2).... 3) ... - necessary info to put into practice by multipliers and decision makers: 
we agree with the reviewer that is important to specify the setting; however,  in order to adhere to the word count required by the journal we decided to indicate only the pre and in service setting which implies the academy education and the training health professional should periodically attend when already employed
> note [page 1]: to me there seems to be no additional info considering the heading: Knowledge ...vegetarian diet ... since mediterranean diet is clearly NOT vegetarian or vegan at all. but maybe better than a conventional mixed diet, however, this phrase seems unnecessarily contribute to the topic, especially in the context with the "bridge" to vegetarian diets and animal welfare/ethical issues - sounds a bit filling lines without deliverancy of content within the first 6-8 lines, which should sell! According to your suggestion, we have deleted this part. Please see the reviewed version.
> highlight [page 2]: 92 First, a literature review was conducted to collect existing information regarding health 93 professionals’ knowledge regarding vegetarian diets. Because no results were retrieved from the available literature: According to your suggestion, we have deleted the sentence
> note [page 2]: It seems that 1-2 sentences lacking considering vegetarian and vegan diets serve as (most) effective tools for some (chronic) diseases, but especially in overweight/obesity, which could markedly contribute to public health of nations. 
We added 1 sentence regarding the prevention of NCD trough vegetarian diet.
> note [page 2]: what about the recruitment, where (Nation? which hospitals, other?), compliance, exclusion criteria only non-medical staff? – according to your suggestion we have given more details on the recruitment.
> note [page 2]: It feels more like a fundamental basic to create a study design and write this paper, than being part of the methods instead, as this is 1st step in starting research: check what is there in literature - maybe better to delete it here and spend 1 short sentence of the lack of this issue in literature in introduction -> to justify the study itself! 
According to your suggestion, we have deleted the sentence and added a short sentence of the lack of the issue in the introduction section.

> note [page 3]: PRIOR to the concrete intervention for the study itself, I guess- mention this somewhere here, as this was necessary to test and rate the quality of your instrument. According to your suggestion, we have specified in the text that the questionnaire was tested before the start of the study
> note [page 3]: Again: compliance as sentence, scheme etc. seems necessary 
We used item as statements affirmations (positive/negative forms) to minimize unintended context effects and maximize the reliability and validity of participants’ responses. Please, see the supplementary material reported at the end of the paper.
> note [page 3]: what about control questions? and to control what item/s? 
We used item as statements affirmations (positive/negative forms) to minimize unintended context effects and maximize the reliability and validity of participants’ responses. Please, see the supplementary material reported at the end of the paper.
> note [page 3]: correct termini Percentage. Done
> note [page 3]: did you test the main effects cobsindering veget. nutritoin knowledge, such as gender or educational degree, rural or urban area of location? 
Yes, we have considered all the variables but we did not find any significant association (please see the result section)With regard to the area of location, all participants worked in a urban area, that is Milan (please see methods section). MANOVA -? We did not perform MANOVADid you formulate a hypothesis from literature review - as this could be nicely be supported to test this hypthesis, or not (why)? Considering the increasing diffusion of vegetarian diets and the potential health benefits associated with their adoption, the primary aim of the study was to assess the health professionals’ knowledge
> note [page 3]: classification is basics for study design, not a method of statistics, remove from here to correct place, maybe should be adressed earlier in Method or even in Intro-Section to better sell the intro to the reader. 
According to your suggestion, we have removed this part from here and put earlier in methods.
> note [page 3]: densify both these phrases (=repetition) 
Thank you for your comment. We modified the phrases.
> note [page 3]: why not the current PP of the AND 2015/2016??? They read the Position of the Academy of Nutrition and Dietetics: Vegetarian Diets. We prefer to use the classification of Sabatè. ..., as semi-vegetarian or pesco-vegetarian each per se is not generally accepted as standing-alone kind of vegetarian diet since fish = meat from fish -? We understand your comments, but in our experience, within the Italian cultural context, this cluster (semi-vegetarian or pesco-vegetarian) is actually very common and better reflect the current diteary habits adopted by people leaving in Italy. Hence, we chose to use this classification.
> note [page 4]: what does years mean in this context, number is missing then? 
Sorry, these is refuse.
> highlight [page 4]: frequencies percentage We change this word and use only percentage. 
Done 
> note [page 4]: no attributes or emotions in the results, sober facts only -> see discussion for interpretation of results! 
We modified the sentence to consider this point
> highlight [page 4]: Remarkbly 
We modified the sentence
> note [page 4]: maybe better as landscape format for better overview of results. 
We have chosen a vertical format to permit an easy and continuous reading; we have made some changes to the table itself in order to improve readability

> note [page 4]: formal presentation of table? The table has been formatted according to the editorial guidelines
> note [page 4]: 7 people from which OTHER nations, would be intersting to know -> 1 sentence within the text 
Thank you for your suggestion, however, since the non-italian subjects were only 7 (1.7% of the entire enrolled population)  we did not keep a record of the specific country of origin.
> note [page 6]: if you chose Le & sabate 2014 for classification, why THEN you pool all veggie kinds you listed among 1 kind of vegetaria diet only? does not seem to make sense -> see AND 2015/2016 Yea, you are right
. We put all veggie kinds together to perform statistical analysis considering subgroups with comparable numbers of subjects included (statistical significance)
> note [page 6]: ... the best of our ... 
We changed the beginning of the sentence, as you suggested (Of the best of our knowledge)
> note [page 6]: formal presentation of tables could be much improved 
The tables have been formatted according to the editorial guidelines
> note [page 6]: for non-experts considering veggie diets it is confusion definition as not clear and strict as eg. AND definition. 
According to your suggestion, we have separated the two terms and given  the specific definitions.

> note [page 7]: Clearly outline the limitations, not as part of another sentence, and there are more than just 2 limitations. We have modified this part,including more limitations and clearly outlining  them. 

> note [page 7]: the effects that influence the lack or extent of nutritional knowledge of health care staff would be highly interesting, is it education, is it self-motivation, it it professional behaviour, is is ... what has the most impact .. add this missing analysis as statistical method is a bit too simple, then as a consequence you have a longer and higher quality discussion, I guess, as discussion section should be 2-3times the length than introduction. so actually it is too short! 

We thank you the reviewer for the suggestion. We actually assessed potential relationship with the collected variables but we could not find any significant association. We have rephrased the conclusion to limit repetitions and increased the length of paragraph
> note [page 7]: repetition! 
We modified this point

Round  2

Reviewer 1 Report

I thank the authors for making appropriate changes and commend them on a well written manuscript. 

I would still appreciate seeing minor revisions to the conclusion as the document remains supportive of the idea that more frequent education would improve the knowledge of these health care professionals when the authors data do not support this. Please let your results guide your conclusions rather than trying to force your data to fit an idea you have molded them to. 

I strongly believe that your data supports a dramatic need for improvements to your education program for all healthcare professionals as the current system does not ensure that these professionals understand basic concepts of nutrition. This is supported by the fact that even those who have been educated within the last 5 years do not have greater knowledge than those who have not. Continuing to utilize this same education program will not improve the outcomes if it has not provided these professionals with the appropriate knowledge in the first instance. 

Author Response

Reviewer 1

I thank the authors for making appropriate changes and commend them on a well written manuscript. 

I would still appreciate seeing minor revisions to the conclusion as the document remains supportive of the idea that more frequent education would improve the knowledge of these health care professionals when the authors data do not support this. Please let your results guide your conclusions rather than trying to force your data to fit an idea you have molded them to. 

I strongly believe that your data supports a dramatic need for improvements to your education program for all healthcare professionals as the current system does not ensure that these professionals understand basic concepts of nutrition. This is supported by the fact that even those who have been educated within the last 5 years do not have greater knowledge than those who have not. Continuing to utilize this same education program will not improve the outcomes if it has not provided these professionals with the appropriate knowledge in the first instance. 

We thank you the reviewer for the suggestion. Accordingly, we have modified the conclusion in the text and in the abstract as follows:

Pre- and in-service education programs should be improved to ensure adequate knowledge of vegetarian nutrition, thus enabling health professionals to provide appropriate educational intervention and guidance, and detect nutritional imbalances that, if not identified in a timely manner, can lead to serious consequences.

 Improving pre- and in-service learning opportunities in vegetarian nutrition for health professionals is strongly advisable.

 Reviewer 3 Report

the paper did improve, please incorporate some minor comments

Author Response

Reviewer 2

Deficient only after checked blood levels, I guess you mean the critical nutrients, not deficient!
This is misleading, since supplementation is only advised when a deficiency is detected, which can be easily avoided by careful mean planning. However, organizations agree for Vitamin B12 to be supplemtented, but here also by vegans only!
Find a more precise wording for this phrase.

We thank you the reviewer for the suggestion. We have modified the text as follows:

The Academy of Nutrition and Dietetics [9], Canada’s Food Guide, the American Dietetic Association, Dietitians of Canada, the American Academy of Paediatrics and the Canadian Paediatric Society has taken favourable position regarding vegetarian diets at all stages of life, including pregnancy, lactation, infancy, childhood and adolescence

as long as appropriate attention to critical nutrients is paid.

What is "High powered" what do you want to say ... is this English understandable - I do not understand the meaning!

We have modified the text as follows: 

probably because the study lacked adequate statistical power to find it